

# Evolution of parasitism and mutualism between filamentous phage M13 and *Escherichia coli*

Jason W. Shapiro[1,2], Elizabeth S.C.P. Williams[1] and Paul E. Turner[1]

[1] Department of Ecology and Evolutionary Biology, Yale University, New Haven, CT, United States
[2] Department of Biology, Loyola University Chicago, Chicago, IL, United States

## ABSTRACT

**Background.** How host-symbiont interactions coevolve between mutualism and parasitism depends on the ecology of the system and on the genetic and physiological constraints of the organisms involved. Theory often predicts that greater reliance on horizontal transmission favors increased costs of infection and may result in more virulent parasites or less beneficial mutualists. We set out to understand transitions between parasitism and mutualism by evolving the filamentous bacteriophage M13 and its host *Escherichia coli*.

**Results.** The effect of phage M13 on bacterial fitness depends on the growth environment, and initial assays revealed that infected bacteria reproduce faster and to higher density than uninfected bacteria in 96-well microplates. These data suggested that M13 is, in fact, a facultative mutualist of *E. coli*. We then allowed *E. coli* and M13 to evolve in replicated environments, which varied in the relative opportunity for horizontal and vertical transmission of phage in order to assess the evolutionary stability of this mutualism. After 20 experimental passages, infected bacteria from treatments with both vertical and horizontal transmission of phage had evolved the fastest growth rates. At the same time, phage from these treatments no longer benefited the ancestral bacteria.

**Conclusions.** These data suggest a positive correlation between the positive effects of M13 on *E. coli* hosts from the same culture and the negative effects of the same phage toward the ancestral bacterial genotype. The results also expose flaws in applying concepts from the virulence-transmission tradeoff hypothesis to mutualism evolution. We discuss the data in the context of more recent theory on how horizontal transmission affects mutualisms and explore how these effects influence phages encoding virulence factors in pathogenic bacteria.

Corresponding author
Jason W. Shapiro, jshapiro2@luc.edu

## INTRODUCTION

A microbial symbiont is a microorganism that depends on a host for completion of at least a portion of its life cycle. Given this broad definition, symbionts may be helpful (mutualistic) or harmful (parasitic) to their hosts. Further, evidence from natural symbiont populations indicates that beneficial symbionts can evolve from harmful progenitors, and

vice versa (*Weeks et al., 2007*; *Sachs, Ehinger & Simms, 2010*; *Sachs, Skophammer & Regus, 2011*). To fully comprehend how symbiotic interactions evolve, it is necessary to consider the evolution of both parasites and mutualists.

Though discussion of mutualists and parasites often occurs separately in the scientific literature, all host-symbiont interactions may produce some benefits and some costs to host fitness (*Bronstein, 2001*). Thus, a mutualist is a symbiont whose benefits to the host exceed any costs, whereas a parasite has the opposite cost-benefit relationship. As such, any change to the selective environment that shifts selection on the benefits and costs of the interaction may also drive an evolutionary transition from parasitism to mutualism or from mutualism to parasitism.

In both the parasitism and mutualism literatures, the relative opportunity for horizontal versus vertical transmission is thought to play an important role in the evolution of symbiont effects on host fitness (*May & Anderson, 1983*; *Ewald, 1987*; *Bull, Molineux & Rice, 1991*; *Bull, 1994*; *Ebert, 1994*; *Day, 2001*; *Wilkinson & Sherratt, 2001*; *Ferdy & Godelle, 2005*; *Foster & Wenseleers, 2006*; *Shapiro & Turner, 2014*). Horizontal transmission occurs when a symbiont enters a new host either through direct contact with an unrelated, infected host, or through contact with a free symbiont in the host's environment. Both parasites and mutualists can be horizontally transmitted. In contrast, vertical transmission is the direct intergenerational transfer of a symbiont from an infected parent to its offspring during reproduction. The virulence-transmission tradeoff hypothesis postulates that when symbiont fitness depends more on horizontal than on vertical transmission, negative effects on current host fitness will have minimal impact on symbiont fitness, and the average cost of infection to the host (virulence) may increase (*Ewald, 1987*; *Bull, 1994*; *Read, 1994*). Alternatively, if symbiont fitness depends more on vertical transmission to host offspring, then symbiont fitness and host fitness are directly linked, and this covariance should select for reduced costs of the symbiont to its host, and potentially for greater benefits, if possible (*Axelrod & Hamilton, 1981*; *Doebeli & Knowlton, 1998*; *Herre et al., 1999*; *Foster & Wenseleers, 2006*). More recent theory suggests that horizontal transmission can also accelerate the evolution of mutualistic symbioses if there exists a mechanism that creates a positive correlation between horizontal transmission and the benefits that symbionts provide to their hosts (*Shapiro & Turner, 2014*). This theory suggests that while one might still expect horizontal transmission to favor the evolution of greater virulence in parasites, its effects on the evolution of mutualism are more difficult to predict and will depend on the mechanisms underlying the interaction.

## Experimental system

Although viruses are often assumed to be strictly parasitic, increasing evidence shows that some viruses benefit their hosts. Examples of mutualistic viruses include *Curvalaria* thermal tolerance virus, which enables Yellowstone panic grass to survive in soils at high temperatures (*Marquez et al., 2007*), the polydnaviruses of parasitoid wasps that suppress the immune systems of other insects to allow the survival of wasp eggs (*Summers & Dibhajj, 1995*), and bacteriophage CTXΦ, which enables *Vibrio cholerae* bacteria to infect the guts of humans and other animals (*Waldor & Mekalanos, 1996*). CTXΦ is a filamentous phage

in the single-stranded DNA virus family *Inoviridae*, which contains other viruses that may enhance bacterial pathogenicity (e.g., Pf4 in *Pseudomonas aeruginosa* (*Rice et al., 2009*) and CUSΦ in both *Escherichia coli* O18:K1:H7 and *Yersinia pestis* (*Gonzalez et al., 2002*)). Temperate bacteriophages, which go through cycles of dormancy and lytic activity, may also be facultative mutualists, as these viruses can carry genes, such as for antibiotic resistance (e.g., *Mazaheri Nezhad Fard, Barton & Heuzenroeder, 2011*) or virulence factors (*Herold, Karch & Schmidt, 2004*), that benefit their hosts in certain environments.

Here, we use the filamentous phage M13 (family *Inoviridae*) and its host, *E. coli*, as an empirical model for studying evolutionary transitions between parasitism and mutualism in response to altered dependence on vertical and horizontal modes of transmission. Filamentous phages provide a powerful model for studying the evolution of symbiosis, because these viruses infect a bacterial cell and produce thousands of progeny viruses without killing the host (*Marvin & Hohn, 1969*). Further, daughter cells inherit the infection from their parent, such that M13 experiences vertical transmission. Over several generations, uninfected daughter cells also arise through the stochastic loss of phage replicative form DNA. Though infected cells are resistant to superinfection (infection by another M13 virion), these uninfected bacterial cells can generally be reinfected (*Merriam, 1977*; *Lerner & Model, 1981*). Thus, M13 naturally undergoes both vertical and horizontal transmission.

Prior work has shown that filamentous phages carrying antibiotic resistance genes could evolve through serial transfer to enhance the benefits they provide bacteria (*Bull, Molineux & Rice, 1991*; *Bull & Molineux, 1992*). In the present study, we set out to test if benefits of phage infection could arise *de novo*. Many proteins encoded by M13 and related phages have pleiotropic effects on host phenotypes, with particularly dramatic impacts on the permeability of the bacterial outer membrane (*Boeke, Model & Zinder, 1982*; *Marciano, Russel & Simon, 2001*). Though we do not directly test for the effects of any particular protein, we suspected that these pleiotropic effects may create avenues for mutualism evolution.

## Experimental overview

To test how transmission mode affects the evolution of symbiont effects on host fitness, we experimentally passaged M13 and *E. coli* in four replicated environments that differed in the availability of naïve, susceptible hosts (i.e., opportunity for horizontal transmission) at the start of each experimental passage (Fig. 1 diagrams the differences among treatments). We will refer to these treatments as "No Host Addition" (N), "Low Host Addition" (L), "Medium Host Addition" (M), and "Complete Host Addition" (C). We also included an "Uninfected" (U) *E. coli* control. The first four treatments represent a gradient from rare horizontal transmission events to regular horizontal transmission at each passage. Further, coevolution may also have taken place in treatments where both M13 and *E. coli* could evolve (N, L, and M). While our data will address this possibility, and we will discuss it in more detail in the Discussion, we will refrain from invoking the terms "coevolve" and "coevolution" through most of the text. Instead, we will describe M13 and *E. coli* that evolved in the same experimental replicate as having been "co-resident", as this term

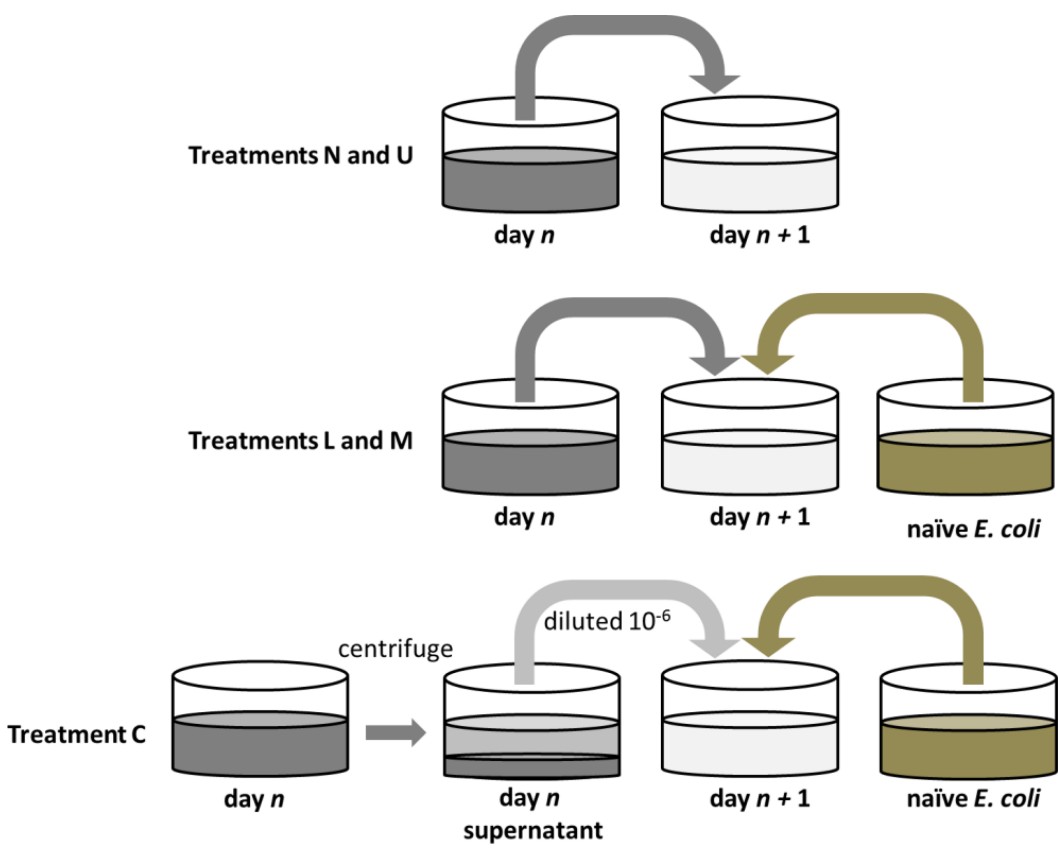

**Figure 1  Diagram of passaging protocols.** In each of treatments N, L, M, and U the day $n$ culture is diluted directly by a factor of 1:$10^2$ into fresh medium for day $n+1$. Uninfected ancestral *E. coli* ("naïve" *E. coli*) is grown in the same day $n$ plate and is added as a supplement of susceptible hosts in treatments L and M. These naïve bacteria make up ∼1% and ∼50% of the initial bacterial density in day $n+1$ wells for treatments L and M, respectively. In treatment C, the day $n$ culture is centrifuged, and the supernatant is diluted by 1:$10^6$ before being combined with naïve bacteria in the day $n+1$ plate.

accurately captures their shared evolutionary history without assuming that evolution by either species is the result of reciprocal evolution by the other.

The cultures were grown in 96-well microplates in a plate reader in order to create a large number of replicate populations per treatment. Unexpectedly, we found that ancestral M13 increased both bacterial growth rate and bacterial density, indicating that the phage is, in fact, a mutualist of *E. coli* in the plate reader environment. After 20 serial passages, we assayed the growth rates and maximum densities of evolved bacteria infected with their co-resident phage. We also assayed these growth traits for the original *E. coli* ancestor when infected with the same, evolved phages. Contrary to common expectations, high rates of horizontal transmission did not lead to a reduction or loss of the mutualism between co-resident phage and bacteria. In fact, infected bacteria from treatments allowing both phage transmission modes had greater improvement in growth rate than bacteria that evolved in the absence of phage. At the same time, phage from these treatments evolved traits that reduced the fitness of the ancestral *E. coli* strain. We discuss these results in light

of recent theory on the evolution of mutualism and explore the relevance of our findings for phage contributions to the evolution of virulence in pathogenic bacteria.

## MATERIALS & METHODS

### Strains of bacteria and phage

*E. coli* CSH22 ($\lambda^-$, *thi*$^-$, *trpR*$^-$, $\Delta$ *(lac-pro)*) is an *F′* K12-derived strain from the Cold Spring Harbor Laboratory collection, notable for a *lac-pro* deletion on the host chromosome that is complemented by a *lac-pro* insertion on the *F′* plasmid (F128). The *F′* plasmid is therefore required for host growth in minimal medium lacking proline. CSH22 was obtained from the Coli Genetic Stock Center at Yale University. Strain JS002 was created by transforming CSH22 with the plasmid *pJS001* (construction described below).

A high-titer stock of phage M13 was generously provided by K. Brooks Low at Yale University. All experiments were initiated from a plaque-purified clone of M13 which was sequenced and found to be identical to the published wild-type "Rutgers" strain (GenBank: JX412914).

### Plasmid construction

Expression of the channel protein, g4p, of filamentous phages is the strongest known inducer of the phage shock operon in *E. coli*, of which *pspA* is a negative regulator (*Model, Jovanovic & Dworkin, 1997*). We obtained the plasmid pDsRedExpress-2 (Clontech), which contains the red fluorescent protein, *DsRedExpress-2* downstream of a *lac* promoter. We amplified the *pspA* promoter region (*Dworkin, Jovanovic & Model, 1997*) from CSH22 using primers SpspAFW (AACC AGCGGAAGAGCTAGCGAGTTCATCAAG) and HpspARV (CTAAAAGCTTCCAGTTTCTGTGGATCTTCC), which also inserted SapI and HindIII restriction sites at the 5′ and 3′ ends of the *pspA* promoter (underlined). We then excised the *lac* promoter from pDsRedExpress-2 by digestion with SapI and HindIII (New England Biolabs) and inserted the *pspA* promoter using T4 DNA ligase (New England Biolabs). The final plasmid is designated *pJS001* and hosts bearing this plasmid express red fluorescence when infected by filamentous phages.

### Growth medium

All cultures were grown in Davis Minimal (DM) medium supplemented to 0.1% with glucose (often called DM1000). Cultures of JS002 were supplemented to 100 µg/ml with ampicillin when grown overnight to ensure retention of *pJS001*.

### Serial passage protocol and freezer storage

All experimental evolution was carried out in BD Falcon 96-well microplates with lids (#351172) in a Tecan F400 plate reader at 37 °C. The microplate was shaken orbitally for 30 s every 10 min (to maintain aeration and to help prevent cell clumping), followed by 2 s of linear shaking (to improve uniform cell suspension) and measurement of absorbance at 600 nm ($OD_{600}$). Continuous shaking was avoided to reduce the probability of hardware malfunction and well-to-well contamination.

The experiment was split into 5 experimental treatments: No Host Addition (N), Low Host Addition (L), Medium Host Addition (M), Complete Host Addition (C), and

Uninfected (U) (see Fig. 1 for a schematic of the setup). All treatments were replicated in 18 wells and serially transferred for a total of 20 passages. In treatments N, L, M, and C, all wells were initiated by inoculating 2 μl of an M13-infected *E. coli* CSH22 overnight culture in 198 μl of DM1000, whereas treatment U wells were similarly initiated with 2 μl of uninfected *E. coli* CSH22. After 24 h of culture under the conditions described above, 2 μl of each population in treatments N and U were added to 198 μl of fresh medium in a fresh microplate. To manipulate treatment L, 2 μl of each population were added to 196 μl of DM1000 with 2 μl of a 1:100 diluted overnight culture of ancestral *E. coli* CSH22, such that ∼1% of the bacteria were uninfected and susceptible at the start of the passage. Treatment M was manipulated so that 1 μl of the population was added to 198 μl of DM1000 with 1 μl of undiluted overnight *E. coli* CSH22, resulting in ∼50% of the starting bacteria as uninfected and susceptible.

After passaging wells for treatments N, L, M, and U, the microplate was centrifuged for 15 min at 4,000 rpm, and the top 100 μl of each treatment C well were transferred to a fresh 96-well microplate for freezer storage and dilution. Each treatment C population was diluted $10^6$-fold, because preliminary assays indicated that this dilution would maintain a population size of ∼$10^4$ particle-forming units (pfu) per 2 μl at time of transfer. Centrifugation and dilution steps also eliminated infected hosts prior to serial passage. 2 μl of each $10^6$-fold dilution of virus was then added to 196 μl of DM1000 with 2 μl of overnight *E. coli* CSH22 so that 100% of the starting bacteria were uninfected and susceptible.

Prior to each experimental passage, the top 100 μl of each well for treatments N, L, M, and C were transferred to a sterile, polypropylene 96-well microplate (USA Scientific 1830-9600) and mixed with 100 μl of 80% glycerol for storage at −80 °C. Undiluted phage from treatment C were also topped with 100 μl of 80% glycerol and stored at −20 °C in the dilution plate.

### Infecting ancestor *E. coli* with evolved phage

Approximately 1 μl of frozen culture was taken from the Day 20 stock plate for each assayed population by "stabbing" with a pipette tip, and then inoculated directly into 200 μl of fresh medium. The plate was incubated for 24 h as described above, and each well was then filtered (Costar Spin-X centrifuge tubes #9301) to remove bacteria. 2 μl of each phage stock and 2 μl of an overnight culture of CSH22 were then added to 196 μl of DM1000 and incubated in the microplate reader. After 24 h, each well was serially diluted $10^{-5}$ and 50 μl of each dilution were spread on DM1000 agar. Plates were incubated for 48 h at 37 °C to obtain colonies of ancestral CSH22 bacteria infected with evolved phage.

### Growth rate assays

We assayed growth rates of bacteria infected with their co-resident phage, and of the original *E. coli* ancestor infected by these same, evolved phages. Assays of evolved phage in the co-resident and original ancestral host backgrounds were completed within the same 96-well microplate and same experimental block. Populations were randomly distributed across the days (blocks) of the assays, with treatments equally represented across blocks.

For controls, in each assay plate we included uninfected ancestor *E. coli* CSH22 and the same bacteria infected with the ancestral M13.

To obtain growth curves of evolved bacteria infected by their co-resident phage, 1 µl of the corresponding frozen stock was inoculated as described above in triplicate, using a randomized design across a 96-well plate. Virus infections were confirmed in a preliminary spot test assay in which each population was grown, filtered to remove bacteria, and then 2 µl of filtrate were plated on top of an LB agar (0.7%) overlay with 200 µl of ancestral CSH22 bacteria. This assay was also performed to check for potential contamination of populations from treatment U. One population from that treatment tested positive for virus infection and this population was excluded from later assays. Further, 15 of the 18 treatment N populations did not yield phage. As a result, only the three remaining populations could be assayed for the effects of phage on the ancestral *E. coli* (see below).

To obtain growth curves of ancestral bacteria newly-infected with evolved phage, two randomly-chosen infected colonies were individually picked using a pipette tip and added to separate wells. The plate was vortexed briefly, and 2 µl of the wells containing bacteria from infected colonies were added to new wells containing 196 µl of DM1000 and 2 µl of the red-fluorescent assay host, JS002 in order to confirm infection. This indicator host was only used to confirm infections in the newly infected colonies. Growth data are reported from the corresponding wells without the indicator host. The plate was then vortexed briefly and incubated in the plate reader at 37 °C for 24 h, taking measurements of $OD_{600}$ and red fluorescence (Ex: 544, Em: 590) every 10 min. The red fluorescence assay confirmed that colonies were infected in the vast majority of cases. Data from uninfected colonies were excluded from the statistical analyses. In rare cases, infected bacteria did not grow sufficiently within the 24 h timespan of the assay, and these data were excluded as well, as it would not be possible to distinguish between an extended lag phase and slow exponential growth in these instances. As mentioned above, free phage could only be recovered from 3 out of 18 treatment N populations. Of these three phage isolates, data from one population were excluded, because infected bacteria did not grow measurably within the 24 h allowed in the assay.

Because assays of co-resident phage and bacteria were initialized from frozen cultures, whereas assays of ancestral bacteria infected by evolved phage were started from infected colonies, minor differences in growth are expected due to potential differences in prior acclimation to the plate reader environment. The ancestral controls included bacteria started from both conditions and are presented separately in Figs. 3 and 4. This difference in starting condition appears to reduce the difference in final density between the infected and uninfected ancestral controls but did not otherwise appear to affect assay results.

## Data analysis

Exponential growth rates of bacteria were determined for each growth curve in R (*R Core Team, 2015*). The data were first smoothed in a moving window of 3 time points (∼30 min) and the exponential growth rate was then determined for every three consecutive points for all OD values above the baseline of detection by the plate reader and below midlog. Maximum OD was also recorded as the highest value obtained over the course of the 24 h

growth period for each curve. This value was chosen as a proxy for yield, since stationary densities could not be estimated for all curves due to noise in some curves late in growth, as well as slow-growing cultures failing to reach a stationary density after 24 h.

Data from each assay were tested for heteroscedasticity using Levene's Test (*Levene, 1960*; *Sokal & Rohlf, 2012*), implemented using the function leveneTest() in the 'car' package in R (*Fox & Weisberg, 2011*). While heteroscedasticity was not observed in the data from growth of the evolved bacteria with their co-resident phage, significant heteroscedasticity was observed in growth data of ancestral bacteria infected with these same evolved phages (see Table S2 for statistics). These data were also notably non-normal in probability plots (not shown). We used the Box–Cox transformation (*Box & Cox, 1964*; *Sokal & Rohlf, 2012*) (estimated by boxcox() in the package 'MASS' in R; *Venables & Ripley, 2002*) on the pooled growth rate data, as well as on the pooled maximum density data from this second assay. The estimated shape parameters were $\lambda = -0.66667$ and $0.141414$ for the growth rate and maximum density data, respectively. After transforming the data, both growth metrics were notably more normal and homoscedastic. Higher variance among treatment C replicates and lower variance among uninfected ancestral *E. coli* samples appeared to account for the original heteroscedasticity. Test significance was corrected for multiple comparisons using the Bonferroni method (*Sokal & Rohlf, 2012*). Two outliers with exceptionally fast growth (one replicate of the infected ancestor in both assays) were excluded from statistical analyses.

# RESULTS

We carried out five experimental treatments with 18 replicate populations per treatment, serially cultured in 96-well microtiter plates. Three treatments ("No Host Addition" (N), "Low Host Addition" (L), and "Medium Host Addition" (M)) permitted both M13 and *E. coli* to evolve, with addition of either 0%, 1%, or 50% of naïve uninfected ancestral *E. coli* at the start of each passage. In the "Complete Host Addition" (C) treatment, the phage populations were isolated and passaged each day with addition of 100% uninfected ancestral *E. coli*, preventing the possibility for bacterial evolution. "Uninfected" (U) controls consisted of bacteria passaged in the absence of phage.

## Assay overview

After 20 passages, we carried out two assays to determine how the phage and bacteria evolved across the treatments. In the first assay, we obtained growth curves of the evolved bacteria when infected by their co-resident phage (treatments N, L, and M) as well as of the uninfected, evolved bacteria (treatment U). This assay identifies broad changes in the interactions between co-resident bacteria and phage, and the uninfected controls identify the capacity for the bacteria to adapt to the experimental environment in the absence of phage infection. In the second assay, we isolated phage from each treatment in which phage could evolve (N, L, M, and C) and used these phage to infect the ancestral *E. coli*. We then obtained growth curves of these newly infected, ancestral bacteria. This assay identifies the extent to which phage evolution can explain changes in the growth of the infected, co-resident bacteria from the first assay.
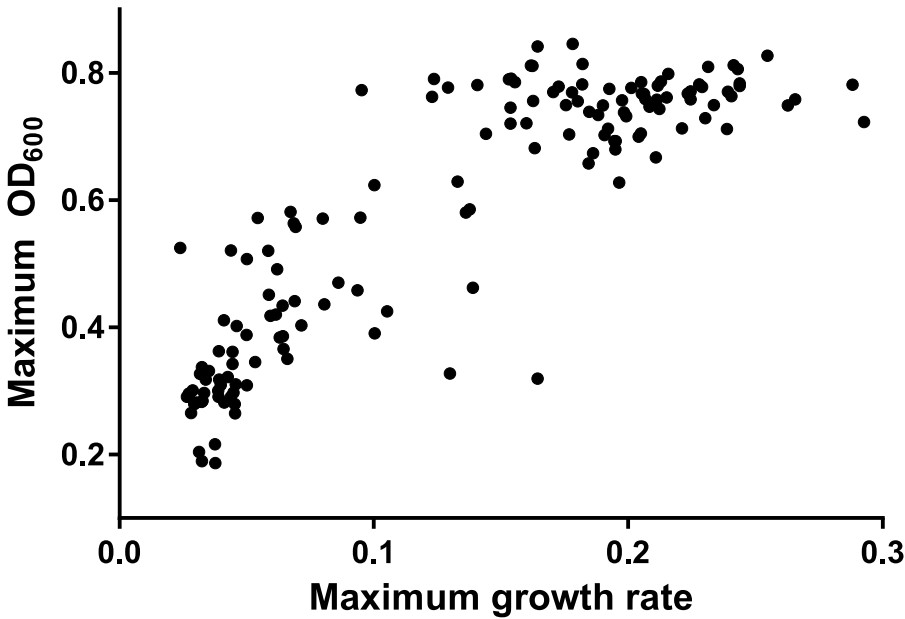

**Figure 2** Scatterplot of maximum OD$_{600}$ against maximum growth rate from all assays.

In each assay, we analyzed the maximum growth rate and the maximum OD$_{600}$ as proxies for the absolute fitness of the bacteria. These two fitness components were strongly, positively correlated ($R^2 = 0.77$ with untransformed data, $p < 0.0001$; see Fig. 2). This positive relationship is important for interpreting the data, as a tradeoff between rate and maximum density would make it difficult to use either metric as a proxy for fitness. Interestingly, there appears to be greater constraint on the evolution of yield than on rate across evolving populations. We will explore this observation in more detail in the 'Discussion.'

### Growth of evolved uninfected bacteria and of evolved bacteria infected with co-resident phage

We obtained triplicate growth curves for each of the evolved bacterial populations infected with their co-resident phage (treatments N, L, and M; see Fig. 3). We also assayed the uninfected ancestor *E. coli* ($n = 6$), the ancestor infected with the ancestral M13 phage ($n = 5$), and the bacteria evolved in the absence of phage (U controls). Results for ancestral strains confirmed that the infected ancestral bacteria grew significantly faster and to significantly higher density than the uninfected ancestor (2-sided $t$ tests, $p < 0.0001$ and $p = 0.007$, respectively), demonstrating that the ancestral infection is mutualistic in the plate reader environment. Results for the U controls showed that uninfected *E. coli* adapted to the plate reader environment, increasing in both growth rate and cell density relative to the uninfected ancestor. While evolved uninfected bacteria also grew to higher maximum densities than the infected ancestral bacteria (2-sided $t$ test, $p < 0.0001$) they did not grow significantly faster than the infected ancestor (2-sided $t$ test, $p = 0.8392$).

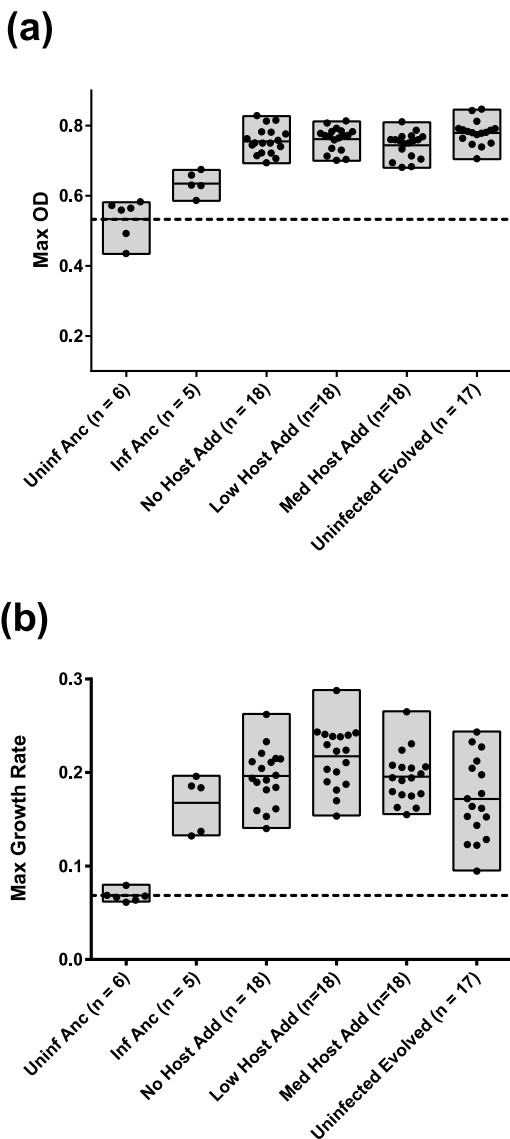

**Figure 3** **Maximum OD$_{600}$ (A) and Maximum growth rate (B) of evolved *E. coli* infected with co-resident phage from each treatment.** The midline in each bar indicates the mean for that treatment. Individual points are the means of triplicate assays for each population. The dashed line depicts the mean peak density (A) or mean growth rate (B) of uninfected, ancestral bacteria.

For both fitness components, there were significant differences among treatments N, L, M, and U, and between these treatments and the infected ancestral bacteria (One-way ANOVAs and $t$-tests summarized in Tables 1–4). On average, bacteria from all treatments evolved a similar maximum density that exceeded that of the uninfected and infected ancestors (see Fig. 3A). This similarity suggested that improvement in maximum density may have been driven by the bacteria alone, and that phage infection did not significantly affect this increase.

We observed greater variation among treatments in the evolution of growth rate. There was significant variation among treatments that started with phage (N, L, and M) and the

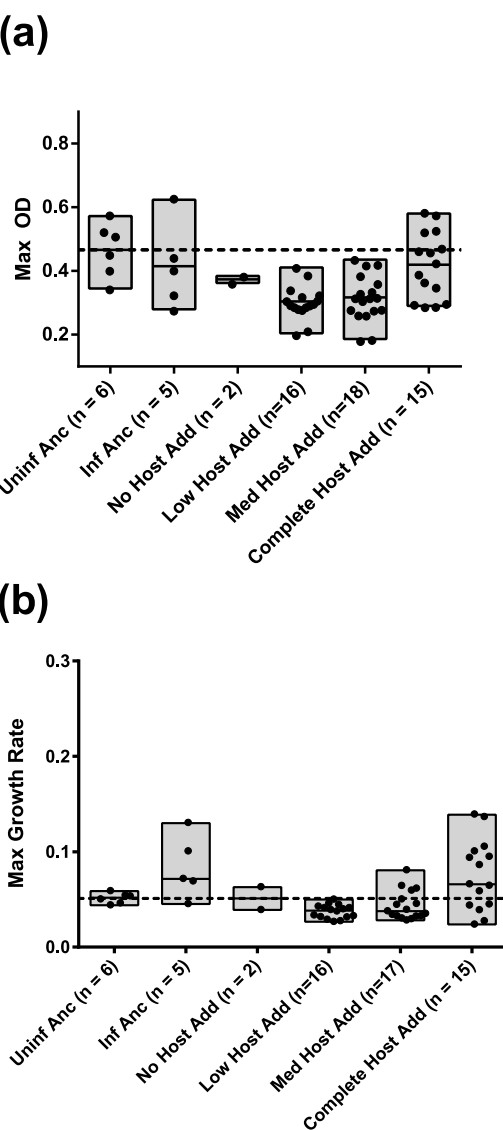

**Figure 4  Growth of ancestral bacteria infected with evolved phage.** Maximum $OD_{600}$ (A) and Maximum growth rate (B) of ancestral *E. coli* infected with evolved phage from each experimental treatment. The midline in each bar indicates the mean for that treatment. Individual points are the means of duplicate assays for each population. The dashed line depicts the mean peak density (A) or mean growth rate (B) of uninfected, ancestral bacteria.

infected ancestral bacteria (One-way ANOVA comparing N, L, M, IA, $p = 0.0141$). Further, this difference is largely driven by evolution in treatment L, which was the only treatment that grew significantly faster than the infected ancestor when tested in pairwise comparisons (2-sided $t$ test of L vs. infected ancestor, $p = 0.0058$). Additionally, Treatments N, L, and M grew significantly faster, as a pooled group, than the uninfected evolved bacteria (2-sided $t$ test, $p = 0.0015$). These data suggest that phage infection affected the evolution of bacterial growth rate. Moreover, there may be an intermediate level of naïve host availability near the 1% used in treatment L that maximizes the evolution of bacterial growth rate.
### Growth of ancestral bacteria infected with evolved phage

We next assayed the growth of ancestral *E. coli*, newly infected with evolved phage from each of the four treatments containing M13 derivatives (i.e., treatments N, L, M, and C). This assay informs the extent to which phage evolution alone is sufficient to explain the results from the previous assay. Pairwise comparisons were not made between data from treatments N, L, and M, as these were not significantly different in either measure by one-way ANOVA. Further, analysis of treatment N phage was limited as free phage could only be recovered from 3 treatment N populations, as noted in the Methods. A possible explanation for the loss of phage production in the 15 remaining treatment N populations is explored in the Discussion.

Surprisingly, bacteria infected with phage from treatment C did not grow significantly differently from either the infected or uninfected, ancestral controls (see Tables 5–8). This result contradicts the common expectation that horizontal transmission may select for increased virulence. Instead, these data suggest that in the plate reader environment, horizontal transmission does not select for virulent M13.

In contrast, while evolved, infected bacteria demonstrated enhanced growth in the first assay, evolved phages from treatments N, L, and M significantly reduced both the growth rate and maximum density of the ancestral *E. coli* relative to infection by the ancestral phage (see Tables 5–8 for statistics). Further, these phage reduced bacterial density below that of the uninfected ancestor, effectively transforming a mutualism into a parasitic interaction (2-sided $t$ test of maximum OD of N, L and M together vs. uninfected ancestor, $p < 0.0001$; see Fig. 4). This finding is particularly unexpected given the results of the first assay, in which phage had no effect on the evolution of co-resident host density. These data suggest that the evolution of phage virulence toward the ancestral host may be a byproduct of local adaptation of M13 to the co-resident, evolving *E. coli* in treatments N, L, and M.

## DISCUSSION

### Mixed transmission modes promote increased benefits to the host

Transmission mode plays multiple roles in the evolution of symbioses. While vertical transmission is generally thought to create partner fidelity feedbacks between host and symbiont, horizontal transmission provides opportunities for symbiont reproduction independent of the original host (*Foster & Wenseleers, 2006*). Here, we tested the effects of differing opportunities for horizontal and vertical transmission on the evolution of a facultatively mutualistic virus and its host bacterium. The virulence-transmission tradeoff hypothesis would have predicted that phage populations experiencing frequent horizontal transmission (treatments C and M) would evolve toward parasitism (reduced mutualism), whereas those with rare or no horizontal transmission (treatments L and N) would maintain or evolve enhanced mutualism. Contrary to this hypothesis, we did not observe the evolution of virulence in treatment C phage. Instead, we found that infected bacteria from treatments with both transmission modes improved in growth, whereas their co-resident phage evolved traits that reduce the fitness of the ancestral host. Further,

only infected bacteria from treatment L grew significantly faster than the infected ancestor, implying a benefit of rare horizontal transmission to the evolution of mutualism.

These data appear to agree with more recent theory that proposed conditions under which relatively rare horizontal transmission is expected to accelerate the evolution of benefits provided by symbionts to hosts (*Shapiro & Turner, 2014*). In order to support this theory rigorously, we would need to uncover a positive covariance between phage horizontal transmission and the mechanism by which phage increase the fitness of infected bacteria. The likeliest mechanism underlying this covariance would be pleiotropy, and it is possible for benefits, costs, and horizontal transmission to all be tied up within the effects of a single gene. For instance, the phage attachment protein, g3p, mediates cell entry, prevents superinfection by related phages, and also increases membrane permeability (*Boeke, Model & Zinder, 1982*). Additional experiments and molecular work would be required to test this hypothesis for the present study.

Alternative explanations for the observed differences in evolved phage effects on co-resident and ancestral bacteria include the possibility that effective phage population size differed among treatments or that phage adapted largely to the differences in the number of ancestral bacteria present across treatments. Though possible, these explanations seem unlikely, as phage in treatment L had the most significant phenotypic evolution, despite likely having lower effective population size and host availability than in treatment C, where no phage evolution was observed.

## The relationship between rate and maximum density across treatments

Figure 2 indicates a strong positive relationship between growth rate and maximum density for the evolving and ancestral populations (both infected and uninfected) in this experiment. Further, the data demonstrate that over the relatively short duration of this study (20 passages), density was constrained to a similar maximum across all treatments, whereas rate varied significantly, with the greatest improvements in rate arising in the populations with phage. These results are consistent with data from long term evolution studies in *E. coli* (*Novak et al., 2006*), in which yield remained mostly constant over thousands of generations while rate continued to improve. The authors of that work, while testing for evolved tradeoffs between rate and yield, also noted that positive correlations between these fitness components would be expected during the earlier stages of adaptation. It would be interesting to continue the current experiment for more generations in order to test if phage infection significantly alters the emergence of a tradeoff between rate and yield.

## The potential role of coevolution

Coevolution is the reciprocal selection on two lineages over time (*Janzen, 1980*). As noted in the Introduction, coevolution in this sense was possible in treatments N, L, and M. While the assays conducted do not directly test whether or not reciprocal selection took place, they do inform the extent to which both partners evolved in each treatment and the role each played in the evolution of bacterial growth rate and density.

Without evolution of both the bacteria and phage, it would not be possible for viruses from treatments N, L, and M to harm the ancestral host while still helping the co-resident

bacteria. Further, phage evolution alone (treatment C) did not lead to significant changes in viral phenotype, and bacterial evolution alone (treatment U), while increasing growth rate and maximum density, could not match the growth rate of infected, co-resident bacteria. These results suggest that coevolution may have played an important role in driving the observed phenotypic changes.

## Limitations in disentangling transmission mode and coevolution

Using the methods discussed, it is not possible to fully disentangle the effects of transmission mode and coevolution. While additional assays and sequencing might uncover the mechanisms underlying the observed results and the relative roles played by both partners, these methods, too, cannot say precisely which of these selective forces had the greatest impact on the evolution of M13 and *E. coli*. Further, it is not currently possible to cure the bacteria of M13 infection in order to assay evolved bacteria in isolation of their co-resident phage. The likeliest method to succeed would require treating infected bacteria with the antibiotic rifampicin, as rifampicin is known to interfere with filamentous phage DNA replication (*Brutlag, Schekman & Kornberg, 1971*). Unfortunately, rifampicin is also known to cause loss of the F plasmid (*Bazzicalupo & Tocchini-Valentini, 1972*), which the bacteria require for growth in the experimental environment. Thus, chemically curing the bacteria of phage is unlikely to succeed without also eliminating the F plasmid or selecting for bacterial resistance to the antibiotic. The latter would then confound the assay by changing the bacterial genotype.

## The loss of phage in vertically transmitted lineages

Given the observed benefits of the phage in the experimental environment, it may be surprising that no phage are produced after evolution in 15 out of 18 replicates with predominantly vertical transmission (N treatment). While one might expect host resistance to evolve rapidly in response to a parasite, why should hosts lose beneficial viruses under predominantly vertical transmission in our experiments? A similar result was observed in previous work in which M13 was modified to carry an antibiotic resistance gene that benefited its *E. coli* host (*Bull, Molineux & Rice, 1991*; *Bull & Molineux, 1992*). The phage were evolved in treatments that manipulated vertical and horizontal transmission, and phage production was lost under vertical transmission (*Bull & Molineux, 1992*). In these cases, the authors observed deletions in the phage that resulted in their maintenance as "cryptic plasmids" or that phage integrated into the bacterial chromosome (*Bull & Molineux, 1992*).

In the present study, we also observed the maintenance of benefits associated with infection in treatment N populations. One possible explanation is that our phage have also evolved into a sort of "cryptic plasmid" as suggested in these earlier studies. We were unable to recover plasmids using traditional miniprep methods, however, from the corresponding bacteria (*Sambrook, Fritsch & Maniatis, 1989*). We also conducted PCR assays targeting phage genes and performed full genome sequencing of five treatment N bacterial populations, and similarly did not find evidence for virus integration into the bacterial genome. These data suggest the phage were truly lost from these treatment N populations, though further sequencing efforts may be necessary to be entirely certain.

An alternative hypothesis is that phage loss is a natural byproduct of viral life history during vertical transmission. In the ancestral interaction, it is known that initial infection by M13 spurs the production of approximately 30–50 copies of the phage dsDNA within the cell, and this copy number dwindles to around 5–15 per cell after vertical transmission (*Lerner & Model, 1981*). Without mechanisms to guarantee fair segregation of these copies into daughter cells, phage-free cells should arise within a few generations (*Merriam, 1977*; *Lerner & Model, 1981*). Further, phage production per cell is expected to decline in established infections as a result of this lower RF copy number within infected bacteria. Even if beneficial to their hosts, eventual loss of phage could be expected.

To test this "neutral loss hypothesis," we analyzed a discrete-time susceptible-infected (SI) model of the experimental setup in R, including steps for dilution at serial passages. (Model details are available in Supplemental Information.) This model does not include any form of mutation or evolution, but rather is intended as a test of under what parameter space one should expect phage to go extinct in treatment N. We explored the model under a range of phage effects on growth rate, from moderate parasitism to moderate mutualism, and also over a range of intrinsic rates of retention. Ultimately, we find that even in the case of mutualists, phage loss can be expected under rates of retention over 90% (Fig. S1). While this model does not rule out the possibility of phage parasitism and host resistance evolution, it demonstrates that vertical transmission is not sufficient to maintain mutualists without sufficiently high rates of both retention in infected hosts and transmission to new susceptibles. These observations also correspond to model results from studies on conjugative plasmid maintenance over time (*Bergstrom, Lipsitch & Levin, 2000*).

Taken together, this model and previous observations of extinction under vertical transmission (*Merriam, 1977*; *Bull & Molineux, 1992*) suggest the phage's life history may have spurred its own loss in treatment N. In light of the data, this explanation seems likely, as phage are maintained in the L and M treatments, where there is a regular supply of susceptible hosts to maintain the phage population and to allow for the eventual re-infection of co-resident bacteria.

## Host tolerance evolution and implications for disease evolution

Two recent reviews highlighted that host evolution of tolerance can affect the long-term evolution of parasite virulence and may enable transitions from parasitism to mutualism (*Edwards, 2009*; *Oliver, Leather & Cook, 2009*). Specifically, host evolution to tolerate the costs of infection can enable the evolution of symbiont traits that increase both the benefits and costs of infection to the host. In our work, this mechanism would explain how bacteria from treatments L and M improve in both rate and maximum density, despite the evolution of viral traits that are harmful to the ancestral bacteria.

Though this study is focused on the evolution of a beneficial phage, this result has important implications for virulence evolution in a variety of pathogens and in zoonoses, in particular. Just as evolution of a virus on one host type can influence viral phenotypes in other hosts (*Turner et al., 2010*), our data demonstrate how the evolution of tolerance by one host genotype may select for symbiont traits that harm a naïve host. This observation also illustrates why it can be so difficult to identify reservoir hosts for zoonotic diseases:

it is not necessary for a new zoonotic disease agent to have ever been virulent toward its reservoir host. A coevolutionary history between a host and a symbiont of any type (parasite, commensal, or even mutualist) may drive the evolution of an unexpectedly virulent disease in an alternate host genotype.

The data also provide insight into the evolution of filamentous prophages (e.g., Pf4, CTXΦ, and YPFΦ) that contribute to the pathogenicity of their host bacteria (*P. aeruginosa*, *V. cholerae*, and *Y. pestis*) toward humans (*Waldor & Mekalanos, 1996*; *Gonzalez et al., 2002*; *Rice et al., 2009*). Though these phages appear to benefit their host bacteria within the context of the bacteria's mammalian host, it is unclear how these phages affect bacterial fitness in other environments or how these interactions have coevolved over evolutionary time. Our results show that the filamentous phage M13, which has not been associated with *E. coli* pathogenicity, can evolve variation in its effects on *E. coli*, and this variation depends on both the ecology of phage transmission and also on the experimental environment. During local epidemics, phage may coevolve with their bacterial host in a manner that later decreases bacterial fitness outside of the eukaryotic host. It is, therefore, important to consider how alternate environments shape the evolution of related phages that contribute to bacterial pathogenicity.

## CONCLUSIONS

In this study, we tested how transmission mode affects the evolution of parasitism and mutualism in a bacteria-phage interaction. We observed the greatest increases in evolved host fitness when both transmission modes were possible. Further, phage from these treatments evolved traits that directly harmed the ancestral host, suggesting pleiotropy at the basis for both parasitism and mutualism in this system. This result undermines the view that symbioses evolve along a continuum between parasitism and mutualism, and instead suggests that discontinuous jumps may occur rapidly as a result of changing environments or interacting with a naïve partner. This inference may be particularly important when considering the emergence of zoonotic diseases or the evolution of pathogenic bacteria in which phage carry virulence-associated genes.

## AVAILABILITY OF SUPPORTING DATA

Raw growth curves, post-analysis growth rate and maximum density data, all statistical analyses underlying Figs. 2, 3 and 4 and Tables 1–8, and R code for the model of phage loss are available through figshare (doi: 10.6084/m9.figshare.2066064). Raw sequencing reads from five evolved Treatment N populations are available through NCBI's Sequence Read Archive (SRP074196).

**Table 1** One-way ANOVA comparing maximum $OD_{600}$ for evolved, infected bacteria.

| Treatments compared | df | F | p-value |
| --- | --- | --- | --- |
| N, L, M, U, IA | 4 | 16.640 | 1.16e–09[**] |
| N, L, M, U | 3 | 2.971 | 0.0379[*] |
| N, L, M, IA | 3 | 17.610 | 3.91e–08[**] |
| N, L, M | 2 | 1.157 | 0.3220 |

**Notes.**

N, No Host Addition; L, Low Host Addition; M, Medium Host Addition; U, Uninfected Evolved; UA, Uninfected Ancestor; IA, Infected Ancestor.

[*]$p < 0.05$.

[**]$p < 0.01$.

**Table 2** Comparison of maximum $OD_{600}$ by treatment for evolved, infected bacteria.

| Comparison (A vs. B) | Difference (A–B) | t | p-value |
| --- | --- | --- | --- |
| NLMU vs. IA | 0.1247 | 7.3008 | 2.69e–10[**] |
| NLM vs. U | −0.0254 | 2.5559 | 0.0128 |
| NLM vs. IA | 0.1186 | 7.0844 | 2.30e–09[**] |
| N vs. IA | 0.1201 | 6.3696 | 2.58e–06[**] |
| L vs. IA | 0.1269 | 7.5210 | 2.18e–07[**] |
| M vs. IA | 0.1079 | 6.0449 | 5.34e–06[**] |
| U vs. IA | 0.1440 | 8.1821 | 8.22e–08[**] |
| U vs. UA | 0.2456 | 12.4590 | 3.63e–11[**] |
| UA vs. IA | −0.1016 | 3.4437 | 7.35e–03[**] |

**Notes.**

N, No Host Addition; L, Low Host Addition; M, Medium Host Addition; U, Uninfected Evolved; UA, Uninfected Ancestor; IA, Infected Ancestor.

[**]Significant at $\alpha < 0.05$ after Bonferroni correction for multiple comparisons.

**Table 3** One-way ANOVA comparing growth rate for evolved, infected bacteria.

| Treatments compared | df | F | p-value |
| --- | --- | --- | --- |
| N, L, M, U, IA | 4 | 4.784 | 0.0018[**] |
| N, L, M, U | 3 | 5.204 | 0.0027[**] |
| N, L, M, IA | 3 | 3.861 | 0.0141[*] |
| N, L, M | 2 | 2.630 | 0.0819 |

**Notes.**

N, No Host Addition; L, Low Host Addition; M, Medium Host Addition; U, Uninfected Evolved; UA, Uninfected Ancestor; IA, Infected Ancestor.

[*]$p < 0.05$.

[**]$p < 0.01$.

**Table 4  Comparison of growth rate by treatment for evolved, infected bacteria.**

| Comparison (A vs. B) | Difference (A–B) | t | p-value |
|---|---|---|---|
| NLMU vs. IA | 0.1521 | 1.6780 | 0.0976 |
| NLM vs. U | 0.1902 | 3.3065 | 0.0015[**] |
| NLM vs. IA | 0.1977 | 2.4393 | 0.0179 |
| N vs. IA | 0.1700 | 1.8928 | 0.0723 |
| L vs. IA | 0.2634 | 3.0702 | 0.0058[**] |
| M vs. IA | 0.1596 | 2.0027 | 0.0583 |
| U vs. IA | 0.0074 | 0.2056 | 0.8392 |
| U vs. UA | 0.8918 | 5.7697 | 1.00e–05[**] |
| UA vs. IA | −0.8844 | −8.0056 | 2.00e–05[**] |

Notes.

N, No Host Addition; L, Low Host Addition; M, Medium Host Addition; U, Uninfected Evolved; UA, Uninfected Ancestor; IA, Infected Ancestor.

[**]Significant at $\alpha < 0.05$ after Bonferroni correction for multiple comparisons.

**Table 5  One-way ANOVA comparing maximum $OD_{600}$ for infections of ancestral bacteria.**

| Treatments compared | df | F | p-value |
|---|---|---|---|
| N, L, M, C, IA | 4 | 5.155 | 0.0014[**] |
| N, L, M, C, UA | 4 | 7.873 | 4.86e–05[**] |
| N, L, M, IA | 3 | 2.705 | 0.0593 |
| N, L, M, UA | 3 | 7.602 | 0.0004[**] |
| N, L, M, C | 3 | 6.440 | 0.0010[**] |
| N, L, M | 2 | 1.009 | 0.3756 |
| L, M, IA | 2 | 3.424 | 0.0435[*] |
| L, M, UA | 2 | 10.686 | 0.0002[**] |

Notes.

N, No Host Addition; L, Low Host Addition; M, Medium Host Addition; C, Complete Host Addition; UA, Uninfected Ancestor; IA, Infected Ancestor.

[*]$p < 0.05$.

[**]$p < 0.01$.

**Table 6  Comparison of maximum $OD_{600}$ by treatment for infections of ancestral bacteria.**

| Comparison (A vs. B) | Difference (A–B) | t | p-value |
|---|---|---|---|
| NLMC vs. IA | −0.1115 | −1.2295 | 0.1453 |
| NLMC vs. UA | −0.1209 | −3.7191 | 0.0041[**] |
| NLM vs. C | −0.1055 | −3.8222 | 0.0001[**] |
| NLM vs. IA | −0.1425 | −2.5314 | 0.0155 |
| NLM vs. UA | −0.1520 | −4.7197 | 4.84e–05[**] |
| C vs. IA | −0.0370 | −0.1361 | 0.8809 |
| C vs. UA | −0.0465 | −1.1810 | 0.3095 |
| UA vs. IA | −0.0095 | −0.8824 | 0.3795 |

Notes.

N, No Host Addition; L, Low Host Addition; M, Medium Host Addition; C, Complete Host Addition; UA, Uninfected Ancestor; IA, Infected Ancestor.

[**]Significant at $\alpha < 0.05$ after Bonferroni correction for multiple comparisons.

**Table 7** One-way ANOVA comparing growth rate for infections of ancestral bacteria.

| Treatments compared | df | F | p-value |
|---|---|---|---|
| N, L, M, C, IA | 4 | 6.047 | 0.0233[*] |
| N, L, M, C, UA | 4 | 5.946 | 0.0211[*] |
| N, L, M, IA | 3 | 5.881 | 0.0023[**] |
| N, L, M, UA | 3 | 2.463 | 0.0773 |
| N, L, M, C | 3 | 3.851 | 0.0924 |
| N, L, M | 2 | 1.697 | 0.1988 |
| L, M, C | 2 | 6.879 | 0.0024[**] |

Notes.

N, No Host Addition; L, Low Host Addition; M, Medium Host Addition; C, Complete Host Addition; UA, Uninfected Ancestor; IA, Infected Ancestor.

[*]$p < 0.05$.
[**]$p < 0.01$.

**Table 8** Comparison of growth rate by treatment for infections of ancestral bacteria.

| Comparison (A vs. B) | Difference (A–B) | t | p-value |
|---|---|---|---|
| NLMC vs. IA | −0.6719 | −3.2080 | 0.0219 |
| NLMC vs. UA | −0.0763 | −0.7808 | 0.4382 |
| NLM vs. C | −0.4741 | −2.8834 | 0.0014[**] |
| LM vs. IA | −0.8114 | −4.1434 | 0.0006[**] |
| LM vs. UA | −0.2158 | −3.7602 | 0.0618 |
| C vs. IA | −0.3373 | −0.0006 | 0.4619 |
| C vs. UA | 0.2583 | 0.0618 | 0.4870 |
| UA vs. IA | −0.5956 | −2.358 | 0.0311 |

Notes.

N, No Host Addition; L, Low Host Addition; M, Medium Host Addition; C, Complete Host Addition; UA, Uninfected Ancestor; IA, Infected Ancestor.

[**]Significant at $\alpha < 0.05$ after Bonferroni correction for multiple comparisons.

## ACKNOWLEDGEMENTS

The authors are thankful to KB Low for providing wild-type M13 and to S Alonzo, KB Low, D Vasseur, and GP Wagner for useful discussion. We are also grateful to J Bull for suggesting we explore a model of phage loss. We thank S Wielgoss and two anonymous reviewers for helpful comments on the paper.

### Funding

This work was supported by the Yale Graduate Program in Ecology and Evolutionary Biology, by the Yale Institute for Biospheric Studies, and by grants to PET from the US National Science Foundation (grant #DEB-1021243) and the NSF BEACON Center for the Study of Evolution in Action (#13-004443). The funders had no role in study design, data collection and analysis, decision to publish, or preparation of the manuscript.

## Grant Disclosures

The following grant information was disclosed by the authors:
Yale Graduate Program in Ecology and Evolutionary Biology.
Yale Institute for Biospheric Studies.
US National Science Foundation: #DEB-1021243.
NSF BEACON Center for the Study of Evolution in Action: #13-004443.

## Competing Interests

The authors declare there are no competing interests.

## Author Contributions

- Jason W. Shapiro conceived and designed the experiments, performed the experiments, analyzed the data, wrote the paper, prepared figures and/or tables, reviewed drafts of the paper.
- Elizabeth S.C.P. Williams performed the experiments, reviewed drafts of the paper.
- Paul E. Turner conceived and designed the experiments, contributed reagents/materials/analysis tools, wrote the paper, reviewed drafts of the paper.

## DNA Deposition

The following information was supplied regarding the deposition of DNA sequences:
Sequence Read Archive (SRA) accession: SRP074196.

## Data Availability

Figshare 10.6084/m9.figshare.2066064.

## Supplemental Information

Supplemental information for this article can be found online at http://dx.doi.org/10.7717/peerj.2060#supplemental-information.

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
