# Peer review of "Evolution of parasitism and mutualism between filamentous phage M13 and Escherichia coli"

_PeerJ, doi:10.7717/peerj.2060_

## Round 0.1 · original submission · Minor Revisions

Thank you very much for the re-submission of your manuscript. I am pleased to inform you that your manuscript will be accepted for publication as soon as you address the minor concerns raised by the two Reviewers. The new modelling part, which answers the concerns of Reviewer 1 was assessed by Reviewer 2 (a computational evolutionary biologist), who raised no issues, thus I trust the model was constructed properly.

I am looking forward to receive the final version of your work.

Reviewer 1 ·

Basic reporting

No comments

Experimental design

My key issue with the first version of this manuscript was with the death of phage in a high proportion of the predominantly vertical treatment (N). The authors have addressed this issue in the text and with a supporting model. Whilst there arguments in the text are seemingly just, unfortunately, however, I do not have sufficient experience with modelling to appropriately critique the model construction and its validity. Perhaps the other reviewers will be able to address the suitability of the model but if not I would recommend that a mathematical modeler (or someone with expertise in this area) look at this new data to ensure it has been constructed appropriately. If the model construction is sound then I am happy that the authors have addressed my concerns.

Validity of the findings

Please see comments on model above.

Additional comments

Clarifying the biology and transmission mode of M13 was very useful in the introduction.

I would change "speak" to "address" in line 116 of introduction

Reviewer 2 ·

Basic reporting

Basic reporting requirements are met.

Experimental design

The basic experimental design was reasonable but unfortunately generated results that are difficult to interpret.

Validity of the findings

The most interesting finding to my mind is that phage that had low or intermediate horizontal transmission possibilities were more deleterious than phage that transmitted purely horizontally.

This finding is in contrast to typical theory that horizontal transmission should lead to higher virulence than vertical transmission. There are a couple of possible explanations:
One, as the authors point out, is that the vertically transmitted phage are more beneficial in the bacteria with which the phage evolved.
Another explanation, is that the phage are mutualistic after they reduce to the lower copy number in cells.
Still another is that the difference was driven by differences in effective phage population size.
It might be useful to mention more of these caveats.

It is then further noted that populations of infected bacteria evolved faster growth rates with phage than without phage. It is again challenging to make the direct comparison between treatments as the effect could have been driven by the addition of ancestral bacteria to the phage treatments. Admittedly this should have if anything reduced the amount of adaptation in the phage treatments, but again the caveat should be noted.

Additional comments

1) It took me a while to figure out why IA results were different in figure 3 and figure 4. I assume this is because in figure 3 the IA were measured after the phage copy number in the cells had gone down but it would be useful to clarify this a bit further.

2) The lack of consistency between reported statistical comparisons is confusing, as is the comparisons that were chosen in each case.

Line 379 - I would be interested to know what kind of gene is envisioned

---

## Round 0.2 · accepted · Accept

Thank you very much for the thoughtful revision of your manuscript. Congratulations on the publication.